# Using Response Surface for Searching the Nearly Optimal Parameters Combination of the Foam Concrete Muffler

**DOI:** 10.3390/ma15228128

**Published:** 2022-11-16

**Authors:** Teng-Hsuan Lin, Jyhjeng Deng, Yi-Ching Chen

**Affiliations:** 1Department of Environmental Engineering, Da-Yeh University, 168 University Rd., Dacun 515006, Taiwan; 2Department of Information Management, Da-Yeh University, 168 University Rd., Dacun 515006, Taiwan

**Keywords:** foam concrete, loudness sensation, noise emission, response surface, glass fiber

## Abstract

A car muffler is a device to improve car noise emission. Some conventional mufflers use layers of glass fiber as a material to absorb noise. However, filling glass fiber is an environmentally unfriendly work, mainly manually filling with chop strand fiber. This research selected a composite material of glass fiber and foam concrete to replace chop strand fiber to avoid this hazard and maintain the muffler’s good noise reduction performance. A response surface methodology with a two-way factorial experimental design repeated the center point twice is performed. The density of the foamed concrete and the weight of the glass fiber is being considered in order to determine the nearly optimal combination of the values in two factors. The response variable is the loudness sensation in Sone of the noise generated from the muffler. At present, the lowest loudness sensation from the two-way factorial design is 16.6494 Sones, which occurred for a muffler with a formula combination of a density of 0.2 g/cm^3^ and 40 g of glass fiber. The significance of this paper is the presentation of a new application of foam concrete to the green muffler design. To the best of our knowledge, this unique area has never been tackled in the material application of concrete. We have discovered that foam concrete indeed does an excellent job in terms of noise reduction as compared with that of a market muffler.

## 1. Introduction

### 1.1. Background

Human beings delight in harmonic sound. Contrary to those delightful sounds, noise is unwanted. High noise intensities have been associated with numerous health effects in adults, including noise-induced hearing loss, high blood pressure, and even mental diseases such as neurosis [1,2]. Moreover, noise could be a hazard for the fetus and newborn [3]. Noise from an automobile is one of the components of noise pollutions to the environment. A muffler is a device used to reduce the noise created inside the exhaust of an internal combustion engine. Generally, the better a muffler reduces sound, the more back pressure is generated. Backpressure is the pressure difference between the average pressure in the exhaust pipe and the atmospheric pressure. Since the average pressure in the exhaust pipe is always greater than that of the ambient atmosphere, the extra pressure will hinder the exhaust gas coming out of the muffler resulting in a lower performance of the internal combustion (IC) engine and additional gas consumption. Therefore, designing a muffler to achieve maximum noise reduction with minimum back pressure is challenging [4]. There are two types of mufflers, those being reactive mufflers and absorptive mufflers. Reactive mufflers use an internal partition of the chamber to cancel out the noise. Absorptive mufflers use absorption material to absorb the sound as it travels through the material and dissipates as heat. Some research has been carried out on the design of the reactive muffler. For example, Oh and Cha [5] use the Taguchi method and a fractional factorial design to determine which factors are essential in reducing the noise.

The experiments are carried out in software of a computer, not physical ones. It was found that the number of segments of the perforated pipe, the radius of the straight pipe, and the length of the straight pipe were significant in reducing the noise. As for the design of absorptive muffler, it is well known through simulation that the bulk density of the glass fiber and the porosity of the perforated duct has an interaction effect on the transmission loss of the muffler [6]. Under the setup of two microphone technique experiments, in the single-pass perforated absorbing muffler with 8% porosity, high bulk density (200 g/L density) performs better in transmission loss than low bulk density (100 g/L density) through most of the frequency range of noise. However, the effect inverts in the case of 2% porosity. Both cases show that mufflers with glass fiber (no matter high or low bulk density) perform better in transmission loss than those without glass fiber. Thus, it is evident that glass fiber indeed can attenuate the engine noise emission.

Moreover, Cofer et al. [7] show that a straight-through muffler filled with glass fiber can reduce the insertion loss in all the range of engine speed between 1000 and 4000 RPM. It is commonly believed that reactive muffler reduces the specific low frequencies noise while absorptive muffler at the high frequencies noise [8,9,10]. Thus, a hybrid muffler is suggested to embrace the advantage of both reactive muffler and absorptive muffler [9,10]. Besides chamber structure and filling material, some researchers have considered using the microperforated plates (MPP) as a covering to enhance the performance of industrial mufflers and car mufflers [10,11,12,13,14]. MPP is thin metal with round holes or slit-shaped holes in the submillimeter range, and its thickness is around 1 mm. When the perforation ratio of holes is larger than 1%, it could create a sound impedance, effectively absorbing sound waves. It is assumed that microperforated plates could act as both reactive mufflers and absorptive mufflers and reduce the low and high-frequency noise [10]. However, their simulation and experimental results reveal that the transmission loss is only available for noise frequencies below 2000 Hz [10,13,14]. The strength of MPP is lightweight and can withstand high temperatures, which is suitable for car muffler conditions. Moreover, it is environmentally friendly because no pollution is emitted. It is dissimilar to filling material such as glass fiber or rock wool in the absorptive muffler. As time goes by, the glass fiber or rock wool could be discharged from the muffler, causing environmental pollution. Both Swedish company www.sontech.se (accessed on 20 March 2022) and Dutch company https://www.redusone.com/ (accessed on 20 March 2022) have been working on MPP muffler; however, the mockup of car muffler is still not available. The strength and weaknesses of car muffler devices are summarized in Table 1.

Noise reduction is also essential in other disciplines. For example, a residential zone requires strict noise loudness control to protect the resident. Thus, various materials and structures have been developed to enhance noise protection. Conventionally, concrete, stone, mortar, and brick are considered to build the wall to prevent noise. The thicker the wall is, the better the noise control. However, it hampers the usage of the room in the building. It also causes a heavy dead weight to the multi-story building. Fortunately, lightweight concrete has been invented to overcome the dilemma. As its name indicates, the lightweight concrete has a good noise reduction capability and low density (much less than the water density such that the lightweight concrete can float on the water). The lightweight concrete has been widely used in the construction of roofs, walls, interior partitions of houses, and shelter [15,16]. More recently, foamed geopolymer reinforced with GFRP (glass-fiber-reinforced polymer) mesh has been proposed for acoustic and thermal insulation with the advantages of lower densities and environmental benefits [17]. However, it is not being considered as a muffler filling yet. It is considered whether the noise reduction of the muffler could be enhanced by the composite material of lightweight concrete and glass fiber. The apparent strength of this composite material is environmentally friendly. The glass fiber will not be discharged into the ambient atmosphere because it is glued together with the lightweight concrete.

### 1.2. Objectives

Moreover, the muffler’s chamber structure is much simpler, and the filling of lightweight concrete is also much more straightforward than filling the layer of glass fiber. Using a composite of lightweight concrete and glass fiber as a raw material for making filling material of absorptive muffler is an ideal route towards sustainable economic growth through less discharged glass fiber in the ambient atmosphere. However, no research study is yet to be reported related to the composite of lightweight concrete and glass fiber as an alternative raw material resource for the production of absorptive mufflers. This present research aims to study the feasibility of the composite as a raw material for making the filling material in the absorptive muffler. This research also assesses the effectiveness of using a varying density of lightweight concrete and amount of glass fiber on the sound reduction performance in the absorptive muffler.

This research proposes the possibility of replacing the glass fiber with the composite of foam concrete and glass fiber in the absorptive muffler. This approach is more environmentally friendly. A response surface is used to search the near-optimal combination of the parameters in the green muffler design. The paper is organized as follows: Section 1 is the introduction, Section 2 is the literature review, Section 3 contains the experimental method and result, including the data analysis and interpretation, and Section 4 is the conclusion.

## 2. Literature Review

### 2.1. On Muffler

The muffler can be classified into the reactive muffler and absorptive muffler. A reactive muffler uses the phenomenon of destructive interference to reduce noise. For complete destructive interference to occur, a reflected pressure wave of equal amplitude and 180 degrees out of phase must collide with the transmitted pressure wave. This design requires a series of resonating and expansion chambers to reduce the sound pressure level at specific frequencies [4]. On the other hand, an absorptive muffler uses absorption to reduce sound energy. A typical absorptive muffler consists of a straight, circular, and perforated pipe encased in a large steel housing. Between the perforated pipe and the casing is a layer of sound-absorbent material that absorbs some of the pressure pulses [8]. Usually, rock wool (basalt wool) and glass fiber are sound-absorbent materials. However, they are with potential health risks; particularly in Taiwan, some muffler manufacturers prefer use shredded glass fiber as the content of absorbent material for the sake of cost-saving. The operators wear the long sleeves dress while putting the shredded glass fiber into the muffler in the semi-automatic filling machine shop. They usually complain about the stingy feeling of the skin. Therefore, the operations represent a serious hazard to workers. Thus, a need to improve the filling process is necessary.

### 2.2. On Foam Concrete

Foam concrete is lightweight concrete with heat insulation and noise reduction [18,19]. It is formed by mixing the foam with the slurry of cement. In our research, the foam is created by stirring up a foaming agent from Gao-Pu enterprise, a local shop in Taiwan, with a puddle mixer in a bucket. After that, the foam in that bucket is poured into another bucket with a cement slurry. Next, the compound of slurry and foam is stirred again by the same puddle mixer to form the foam concrete. A simple process flow chart of lightweight concrete used in our research is shown in Figure 1. A similar process can be seen in Kumar and Mishra [20]. Note that the unique feature of this process is that the tripartite glass fiber, foam, and slurry are mixed to form fresh, lightweight concrete (LWC). The difference between them lies in the following aspects. First, [20] has a dry mixing procedure before the dry mix is formed into the slurry mix, whereas ours use cement instead of dry mix. Second, we added glass fiber after a fresh LWC mix, whereas [20] did not. The glass fiber is from a local shop in Taiwan, shown in Figure 2. After the foam is made, it is poured out from the red bucket into the blue bucket, where a slurry of cement is shown in Figure 3. Note that the puddle mixer is in the bottom right of Figure 3. Then slurry mix with the proper amount of glass fiber is added to the muffler’s drome. For reference’s sake, the chemical composition and physical properties of the glass fiber from the supplier are provided in Table 2 and Table 3. During the testing of making various densities of foam concrete, four 20 × 20 × 20 cm wooden molds are made to contain the slurry to measure the actual density of the LWC. The foam concrete can be poured into any mold and remain there for two to three days to solidify, as shown in Figure 4. The foam concrete has a high shield capability of heat insulation. In the example of the tourist center in Zhongli Service Area of highway 1 in Taiwan, the shelter made by foam concrete can reduce the room temperature inside the hut by 10 degrees Celsius, according to its construction firm, Young-An mineral technology. However, according to our field test, a one-time measurement by one of the authors, the outdoor temperature is 32 °C, whereas the inside temperature in the shelter is only 26 °C. Figure 5 shows that the shelter is in a curved shape, indicating the foam concrete’s plasticity.

### 2.3. On Sound Absorption

The sound absorption of foam concrete is very controversial in literature. It is mainly due to the different measures of sound absorption. For example, Kim, Jeon, and Lee [19] show no difference in acoustic transmission loss among the various percentages of the air-entraining (AE) agent in the concrete ranging from zero to 1.5% AE. The transmission loss is measured in frequencies from 10 to 3000 Hz. Thus, the transmission loss does not vary with different densities of the lightweight concrete ranging from 1.2–1.8 g/cm^3^, indicating that the density of the lightweight concrete has nothing to do with the sound absorption capability. However, Laukaitis and Fiks [18] show that the foam concrete density is somewhat related to the mean normal incidence absorption coefficient. The lower the density is, the higher the mean normal incidence absorption coefficient becomes, indicating a higher sound absorption capability. Kim, Hong, and Pyo [21] also show that the total porosity of the foam concrete is not completely related to the noise reduction coefficient (NRC) and sound absorption average (SAA). The trend shows that the higher the total porosity is, the higher the NRC and SAA are. However, there is some exception and no linear relationship between total porosity and NRC/SAA. Note that the higher the porosity is, the lower the density of foam concrete. Thus, they cannot conclude that the lower the density of the foam concrete is, the better the sound absorption is.

This research uses a novel approach to replace glass fiber in the absorptive muffler with foam concrete and glass fiber composite. Here, we reveal the thinking process in arriving at this approach. Glass fiber is for the sound absorption of the noise coming from the car’s exhaust. Foam concrete is for the sound absorption from the ambient atmosphere outside the wall. Since composite materials are better than individual materials in sound reduction, can the glass fiber be replaced by the composite of the foam concrete and glass fiber to perform the better function of the sound absorption from the exhaust and render an environmentally friendly working condition for the operators? The answer is definite and encouraging after a few trials runs. The prototype of the foam concrete muffler is tested in the anechoic chamber at the DaYeh university. A sound is generated and passes through the muffler; next, its sinusoidal curve is rendered. The experimental setup is shown in Figure 6. A microphone detects the outgoing sound on the left in Figure 6. The lower the sinusoidal curve is, the better the sound absorption of the muffler is. Since one can set up different parameters of the foam concrete and glass fiber to measure its corresponding sinusoidal curve, one then can know which parameters combinations can perform better in sound absorption.

### 2.4. On Response Surface Methodology

Response surface methodology (RSM) is a collection of tools developed in the 1950s to determine optimum operating conditions in applications in the chemical industry [22]. In this research, RSM is used to determine the optimal combination of foam concrete density and the glass fiber’s weight to reduce the noise. The dependent variable of the RSM is a univariate versus the sinusoidal curve. Thus, one must convert a sinusoidal curve into a single index and regard this as a data reduction problem. Since the sinusoidal curve is a set of data points on sound pressure level in dB scale (dBSPL) along with the frequency domain, one needs to convert the data points into an aggregate number representing the dBSPL of the noise. Previous research has tried to convert the sound into a single index. For example, as we mentioned before, Laukaitis and Fiks [18] used the mean normal incidence absorption coefficient ${\bar{\alpha }}_{0}$ to represent the noise level. Note that “mean” here stands for the estimate of the average of a population of normal incidence absorption coefficient α (Kim and Lee [23]). Note that α is a function of frequency.

Note also, for each experiment, only one observation of the mean normal incidence absorption coefficient ${\bar{\alpha }}_{0}$ was obtained. Linear regression is used to estimate the mean normal incidence absorption coefficient ${\bar{\alpha }}_{0}$ regarding different air permeability or proportion of connected pores in the aerated concrete slabs. Laukaitis and Fiks used ISO 354 standard (https://www.iso.org/standard/34545.html, accessed on 20 March 2022) to calculate the normal incidence absorption coefficient. However, he did not specify the detail. According to Forouharmajd and Mohammadi [24], similar methods with ISO-10534-1 (https://www.iso.org/standard/18603.html, accessed on 20 March 2022) and ISO-10534-2 (https://www.iso.org/standard/22851.html, accessed on 20 March 2022) measure the normal incidence absorption coefficient. A straight impedance tube with two microphones attached to the surface of the tube is used to measure the reverberant noise level in the tube. A VA-lab4 software (version 5.5, https://va-lab4.software.informer.com/, accessed on 20 March 2022) is used to measure the sound absorption coefficient α. Note that α is a value between zero and 1, with 1 indicating complete absorption of the reverberant sound. Note that α varies with the frequencies of the noise. One drawback of the normal incidence absorption coefficient is that α does not increase concerning the thickness of the specimen. This phenomenon is contrary to our natural concept that the thicker the specimen is, the better the sound absorption.

In contrast to the mean normal incidence absorption coefficient used by Laukaitis and Fiks, Kim, Jeon, and Lee [19] use an impedance tube to measure the acoustic transmission loss according to ASTM E2611 (https://www.astm.org/e2611-19.html, accessed on 20 March 2022). The experimental setup is quite similar; however, different measures are taken. The experiment follows ASTM E2611 using four microphones instead of two in the impedance tube as in ISO-10534-1 and ISO-10534-2. The transmission loss (TL) is defined as in a specified frequency band, ten times the common logarithm of the reciprocal of the sound transmission coefficient, which is defined as the fraction of airborne sound power incident on a material that is transmitted by the material and radiated on the other side. The bigger the TL is, the better the sound shielding capacity. Concerning sound transmission loss (STL), interested readers can refer to [25] for further definition and its application with porous and other advanced materials. Note that contrary to a single index in normal incidence absorption coefficient α, the TL is a function of frequency. According to Kim, Jeon, and Lee, the acoustic transmission loss of standard and lightweight concrete with and without air-entraining (AE) agents is no different from each other. The result is contrary to Laukaitis and Fiks. The mean normal incidence absorption coefficient ${\bar{\alpha }}_{0}$ is linearly inverse to the proportion of connected pores. It implies that ${\bar{\alpha }}_{0}$ is linearly proportional to the density of the foam cement concrete.

Kim, Hong, and Pyo [21] use the multi-layered perforated panel model proposed by Maa to simulate the sound absorption of the developed concrete. The acoustic absorption curves are established by measuring the acoustic absorption coefficients under different frequencies. Only the curve’s highest peak and corresponding frequency are recorded to represent the acoustic absorption. As the open void ratio becomes higher, the acoustic absorption improves in general, and the peaks of the curves shift to higher frequencies. Afterward, a single value is condensed from the acoustic absorption curves. Two indexes are provided, those being the noise reduction coefficient (NRC) and the sound absorption average (SAA). The NRC is evaluated as the arithmetic average coefficient at 250, 500, 1000, and 2000 Hz.

In comparison, the SAA is the average arithmetic value of the acoustic absorption coefficient at 12 1/3 octave frequencies ranging from 200 to 2500 Hz according to ASTM C432-17 (https://webstore.ansi.org/Standards/ASTM/astmc42317, accessed on 20 March 2022). Note that if NRC and SAA are higher, the better the acoustic absorption is. In the void ratio experiment of Kim, Hong, and Pyo, neither NRC nor SAA are good predictors of acoustic absorption because they do not indicate that the higher the open void ratio is, the better the acoustic absorption. Thus, a better single value still needs to be sought to represent the acoustic absorption curve.

### 2.5. On Sone

As mentioned above, all the indexes introduced by the three pieces focus on the physical measurement of the sound. Whether mean normal incidence absorption coefficient ${\bar{\alpha }}_{0}$, transmission loss (TL), or NRC and SAA, they all neglect the one crucial part, the hearer’s perception. According to psychoacoustics, the same sound pressure level in different frequencies means different loudness levels [26]. Since noise reduction is for the wellness of human beings, thus it is reasonable to consider the perception of the sound pressure level by human beings. In our research, we conduct a series of foam cement mufflers on the anechoic chamber to investigate the proper parameters in the foam cement so that the muffler will render the most negligible noise. Thus, the loudness sensation (Fastl and Zwicker [26]) is used to measure the quality of the noise reduction of the muffler. The unit in the loudness sensation is Sone. When the sound pressure level is 40 dB in the 1 k Hz pure sound, its loudness sensation is one Sone. Equal loudness contour in Sone is shown in Figure 7 [27]. There is another measure of loudness level in Phon where the sound pressure level in dB in the pure tone of 1 k Hz is treated as the loudness level in Phon. Thus, Figure 7 shows that the contour line of 40 Phons where the sound pressure level in the pure tone of 1 k Hz is 40 dB, while the loudness sensation in the pure tone of 1 k Hz is 1 Sone.

However, most of the sound is not pure tone. Instead, it is a combination of sound in different frequencies, especially in the noise from the muffler. Thus, Kook et al. [28] proposed a process to convert a sound with a sound pressure level curve in frequency to a specific loudness curve in critical band rate. Next, an integration in the calculus of the particular loudness curve renders the loudness sensation in Sone. In our research, the Kook et al. approach is applied to find the noise level of the green muffler in the anechoic chamber.

## 3. Method and Result

### 3.1. Two-Way Factorial Design

A two-way factorial design (Vardeman [29]) with the center point being repeated is performed to construct the response surface (Myers, Khuri, Carter [22]) to find the optimum combination of parameters such that minimum loudness sensation may be reached. The purpose of repetition on the center point is to estimate the variation of the statistical model. Ideally, it is better to repeat the two-way factorial twice; however, it is time-consuming and costly. Thus, the only center point is repeated twice. The two factors are the foam cement concrete’s density and the glass fiber’s weight. The experiment layout and result are shown in Table 4.

Table 4 is explained as follows. First, the Density means the density of the foam cement concrete, and Weight means the weight of the glass fiber added to the green muffler. One material cannot block out all the noise in different frequencies; however, the composite material can reduce most of the noise in different frequencies (Tang and Yan [30]). Porous concrete that uses a composite of expanded perlite and slag as aggregate can obtain good acoustical absorption properties and satisfy mechanical requirements (Zhao, Wang, Wang, and Liu [31]). In our experiment, two kinds of sound pressure level curves are extracted. One is directly from the anechoic chamber; the other is from the noise lined out to the notebook. The latter is used for the loudness sensation in our experiment because the former cannot be retrieved and stored. A schematic drawing is shown in Figure 8. Figure 8 shows that inside the anechoic chamber, there is a sound generator on the right side of the green muffler, and a sound detector is on its left side. Then the sound is piped out to the analyzer to derive its sound pressure level curve and displayed on the PC screen. At the same time, the sound is lined out to the notebook to be heard by the authors. Note that the sound in the lined out will be different (actually smaller) from the sound detected by the sound detector because of the transmission loss; however, the sound pattern of the lined-out sound should be similar to that in the sound detector.

As an illustration, the two sound pressure level curves for density 0.8 g/cm^3^ and fiber 0 g are displayed in Figure 9. One is from the analyzer as shown in red, while the other is from the lined out sound as shown in green, which is converted to its corresponding sound pressure level curve via the accousticLoudness function in Matlab (R2020b, https://www.mathworks.com/products/matlab.html, accessed on 20 March 2022). Note that the red and green curves hold a similar pattern, yet the green curve shows less strength in sound pressure level than the red one. This research uses the lined out sound as our output in the two-way factorial experiment; thus, the loudness sensation is 35.2277 Sones. We can use the response surface to fit the ten data points in Table 4 and acquire Figure 10.

### 3.2. Patch Formula and ANOVA

Note that in Figure 10, there are two centered points marked in red in the middle of the surface. There are four points in red on the four corners around the surface patch. In addition, four midpoints in black are on the boundary of the patch. The patch is in the range of 0 to 40 g in fiber glass weight and 0.2 to 0.8 g/cm^3^ in the density of foam concrete. Using Matlab [32], the formula of that patch is,
Sones = 15.99 + 69.7 Density + 0.270 Fiber − 57.4 Density × Density − 0.01395 Fiber × Fiber + 0.487 Density × Fiber(1)

If we redraw the surface patch in the new broader range of 0 to 60 g in fiber glass weight and 0 to 1.2 g/cm^3^ in the density of foam concrete, we acquire Figure 11. It shows that the minimum point of the patch occurs at a point where weight is 60 g, density is 0.2 g/cm^3^, and the corresponding Sones is −0.5861, which is an impossible value in practicality. No loudness sensation could be lower than zero. This phenomenon shows that the Equation (1) model is incorrect in estimating the minimum point in that region. Equation (1) is a concave function; thus, no global minimum is available. For a concave function such as that one, only a local minimum is available, depending on the region considered. As a reference, its analysis of variance (ANOVA) is shown in Table 5. The *p*-value of the lack of fit for this model in Equation (1) is 0.259 meaning the model does not fit the data well.

### 3.3. Matlab’s Interpolation

We try different models other than response surfaces, such as thin-plate spline interpolation, bi-harmonic interpolation, and linear interpolation in the Matlab modules (Matlab [32]). Their result is shown in Figure 12, Figure 13 and Figure 14. Note that the thin-plate spline interpolation in Figure 12 shows that in the range of 0 to 40 g in fiber glass weight and 0.2 to 0.8 g/cm^3^ in the density of foam concrete, a local minimum point occurring in the point where weight is 37 g and density is 0.2 g/cm^3^, and the corresponding Sones is 12.3269. It also shows the weight of fiber glass is not linearly proportionate to the dependent variable Sones. Furthermore, it indicates that one cannot achieve the lowest loudness sensation by adding the glass fiber into the green muffler. The function of thin-plate spline interpolation is complex, and it is hard to explain the trend of the surface, particularly the narrow ditch of the surface patch nearing the weight of 20 g. The bi-harmonic interpolation operates in the opposite way to the thin-plate spline interpolation. The surface pops up in the weight region as 10 or 30 g in bi-harmonic interpolation, as shown in Figure 13, whereas it sags in the region nearing the weight of 20 g. The local minimum point occurs when weight is 40 g, density is 0.2 g/cm^3^, and the corresponding Sones is 16.6496. As in linear interpolation in Figure 14, we found two promising regions for further investigation, those being optimum region 1 and sub-optimum region 2, as shown in Figure 15.

In the optimum region 1 in Figure 15, two more experiments can be performed at the center point where weight is 30 g, and density is 0.3 g/cm^3^. This way, one can establish a response surface to find its local minimum points. On the other hand, in the sub-optimum region 2 in Figure 15, similar experiments can be performed at the center points where weight is 10 g, and density is 0.3 g/cm^3^.

Through different modeling of the data in Table 4, some light is shed on the possible combination of weight and density to reach a near minimum point in loudness sensation in Sones. One is the lowest data point in Table 4, where weight is 40 g, and density is 0.2 g/cm^3^; its corresponding loudness sensation in Sones is 16.6496. The other is the point where weight is 37 g, density is 0.2 g/cm^3^, and the corresponding estimated Sones is 12.3269, as shown in the thin-plate spline interpolation. Still, there are possible local minimum points in the optimum region 1 and sub-optimum region 2 obtained through further experiment as shown in the linear interpolation.

Now, how good is the performance of loudness sensation in Sones at a point where weight is 40 g, and density is 0.2 g/cm^3^, its corresponding loudness sensation in Sones is 16.6496? Its sound pressure level curve is shown in Figure 16 for reference. To compare its performance, we can measure it against an ordinary muffler from the market. The sound pressure level of a market muffler is shown in Figure 17. Its corresponding loudness sensation in Sones is 14.6347. If we put the sound pressure level curve lined out in Figure 16 and Figure 17 together, it gives Figure 18. In order to demonstrate the effectiveness of the sound absorption, we superimpose Figure 18 with the corresponding sound pressure level from the empty barrel and render Figure 19. The loudness for the empty barrel is 34.6768 Sones. From Figure 19, it is clear that although there is a difference of 2.0149 Sones, the margin only counts 10.58% of their average noise reduction (1/2 × ((34.6768 − 16.6496) + (34.6768 − 14.6347)) = 19.0347 Sones), but the two curves are intermixed together. The sound pressure level curve of the market muffler (in magenta) is only better in the high-frequency range (2000~7000 Kz) than that of density 0.2 g/cm^3^ and fiber 40 g. The noise difference in that range is not detectable for ordinary human beings. It is clear from Figure 19 that the significant reduction in loudness from the market muffler or green muffler is in the region of 200~10,000 Hz. On the contrary, the empty barrel performs better in noise reduction in the low-frequency region of 25~250 Hz. Neither market muffler nor green muffler prevails over the empty barrel in all frequency regions. It is necessary to investigate the phenomenon in future study.

### 3.4. Comparison among Three Mufflers

As a comparison, one imbeds the three sound files in waveform audio file (wav) format in Table 6. These three files are retrieved from the sound coming out from an empty barrel, market muffler, and green muffler, respectively. Their names are empty barrel, market muffler, and green muffler. The empty barrel means that there is no sound insulation in the barrel. The market muffler means the sound file is retrieved from the market muffler. The green muffler with a density of 0.2 g/cm^3^ and fiber 40 g means the sound file is retrieved from the green muffler. As you may hear, there is a big difference between the loudness between the empty barrel, the market muffler, and the green muffler. However, there is a negligible difference between the market muffler and green muffler sound waves. Finally, for completeness, we list the formula of 0.2 g/cm^3^, as shown in Table 7. There are three mixtures in Table 7.

Mixture 1 is for cement slurry. Note that water 1 stands for the first portion of water used in the mixture 1. Part of the cement is replaced by lime and calcium sulfoaluminate (CSA). The water to cement ratio is 0.468. The substitute of Portland cement with CSA refers to ultra-low density foam concrete by Jones, Ozlutas, and Zheng [33]. The second mixture is for foam making. The foaming agent is about 5.5% of the total solution; thus, one uses 38 g of foaming agent and 657 g of water 2. Note that there are 695 g in total for the foam solution. The ratio between mixture 2 to mixture 1 is 695/1693 = 0.41. As the ratio becomes lower, the density of the foam concrete becomes higher. Finally, for mixture 3, there is calcium chloride anhydrous and water 3. Calcium chloride anhydrous is used to dehydrate the water in the foam concrete to harden it quickly. One first mixes mixture 1 into the slurry for 5 min with a spatula. Next, mixture 3 is stirred for 6 min and then poured into mixture 1. Following that, the foam is mixed with a puddle mixer for 7 min for mixture 2. As the next step, the foam is poured into the cement slurry in the mixture and mixed with the puddle mixer for 15 min to render the fresh, lightweight concrete. We produced low density 0.2 g/cm^3^ foam concrete with dimensions of 20 cm × 20 cm × 20 cm three times to confirm the result. The fresh wet densities are 0.197, 0.199, and 0.215 g/cm^3^, respectively. After 25 days of natural curing at room temperature, their densities decrease to 0.172, 0.175, and 0.186 g/cm^3^, respectively. The result of those three slabs after 25 days of curing is shown Figure 20. It is obvious that the shape is well preserved even though their dry densities are below 0.2 g/cm^3^.

## 4. Conclusions

High noise intensities have been associated with numerous health effects in adults and newborn babies. Among many noise sources, cars produce a significant amount of noise, which can be attenuated via mufflers. Thus, the design of an environmentally friendly muffler is a crucial matter. To this end, this research proposes replacing the glass fiber with the composite of foam concrete and glass fiber in the absorptive muffler. A ten-point two-way factorial experimental design is performed, factors being the density of foam concrete and fiber weight. Whereby the center point is repeated twice to estimate the variation of the statistical model. Next, a response surface is used to search the near-optimal combination of the parameters in the green muffler design. The significance of this paper is twofold. First, it replaces the glass fiber with a composite of foam concrete and glass fiber. As time goes by, the glass fiber could be discharged from the muffler, causing environmental pollution. The apparent strength of this composite material is environmentally friendly. The glass fiber will not be released into the ambient atmosphere because it is glued together with the foam concrete. To our best knowledge, this unique area has never been tackled in the material application of concrete. We have discovered that foam concrete indeed does an excellent job in terms of noise reduction as compared with that of a market muffler.

Second, we retrieved the sound from the anechoic chamber and compared different kinds of mufflers. In our research, we allow the audience to hear the noise and show the sinusoidal curves with their corresponding single index, loudness in Sones. This measurement differs from the conventional measurement based only on the physical meaning, whereas ours consider both the physical and psychological sense. Such a presentation is unique, as it engages with audible and visual senses. The loudness in Sone becomes more meaningful to the audience. Previous research used mean normal incidence absorption coefficient ${\bar{\alpha }}_{0}$ to represent the noise level as by Laukaitis and Fiks [18], acoustic transmission loss as by Kim, Jeon, and Lee [19], and the noise reduction coefficient (NRC) and the sound absorption average (SAA) as by Kim, Hong, and Pyo [21]. However, none of those indexes are related to human sensation. They only measure the physical quantity of the sound. However, according to psycho-acoustic, humans sense different loudness not only in sound pressure but also in frequency. Thus, in this research, a different index is taken, the loudness sensation in Sones is used.

Based on the ANOVA of the ten-point experiment, one cannot obtain the global minimum point with the response surface because the surface patch is not a convex function. The function in the surface patch can render an unreasonable point for the local minimum. Moreover, the lack of fit in the ANOVA shows the *p*-value for the model is 0.259, indicating the model does not fit the data well. However, the promising result still can be drawn from such experimental failure. First, a minimum point is obtained in the ten-point experiment, a point at density 0.2 g/cm^3^ and fiber 40 g. The loudness sensation of the lined-out noise is 16.6496 Sones.

Moreover, with the thin-plate spline interpolation model in Matlab, one can obtain a possible local minimum point at weight is 37 g, density is 0.2 g/cm^3^, and the corresponding Sones is 12.3269. The linear interpolation of Matlab shows two promising regions for further investigation. First, in the optimum region 1 in Figure 15, two more experiments can be performed at the center point where weight is 30 g and density is 0.3 g/cm^3^. This way, one may establish a response surface to find its local minimum points. Similar experiments can be performed in the sub-optimum region 2 in Figure 15 at the center points where weight is 10 g, and density is 0.3 g/cm^3^.

We compare the minimum point in the ten-point experiment with the loudness sensation from the muffler in the market; its value is 14.6347 Sones. Although there is a difference of 2.0149 Sones between the two, the margin only counts 10.58% of their average noise reduction, but the two curves are intermixed together, as shown in Figure 18. The sound pressure level curve of the market muffler (in magenta) is only better in the high-frequency range (2000~7000 Kz) than that of density 0.2 g/cm^3^ and fiber 40 g. The noise difference in that range is not detectable for ordinary human beings. Figure 19 also shows both green muffler and market muffler perform better than an empty barrel in the region of 200~10,000 Hz. On the contrary, the empty barrel performs better in noise reduction in the low-frequency region of 25~250 Hz. Neither market muffler nor green muffler prevails over the empty barrel in all frequency regions. Therefore, it is necessary to investigate the phenomenon in a future study. We also list the sound waves of a green muffler, market muffler, and empty barrel in Table 6 to allow the audience to hear their difference. Finally, a detailed working formula for making foam concrete of density 0.2 g/cm^3^ is listed in Table 7.

## Figures and Tables

**Figure 1 materials-15-08128-f001:**
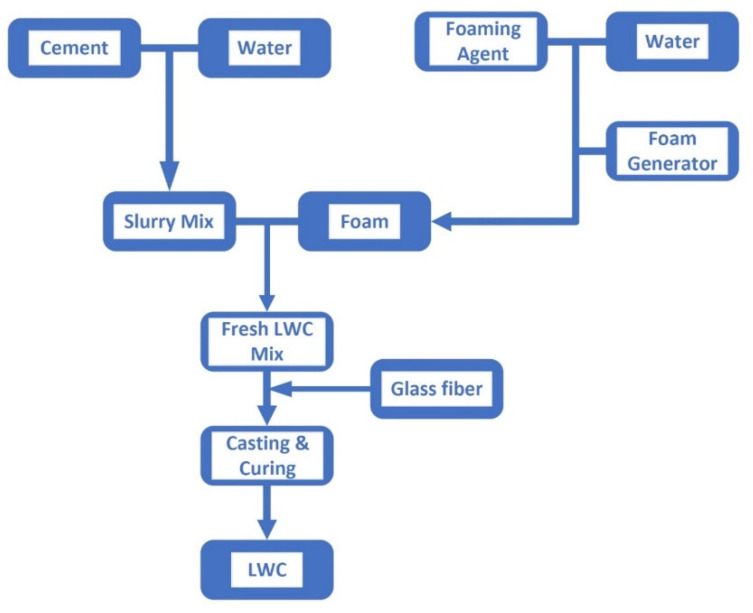
The manufacturing process of light weight concrete (LWC).

**Figure 2 materials-15-08128-f002:**
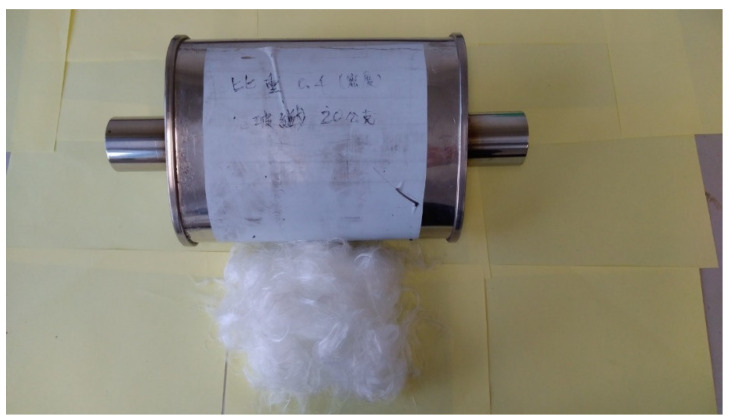
The glass fiber with muffler.

**Figure 3 materials-15-08128-f003:**
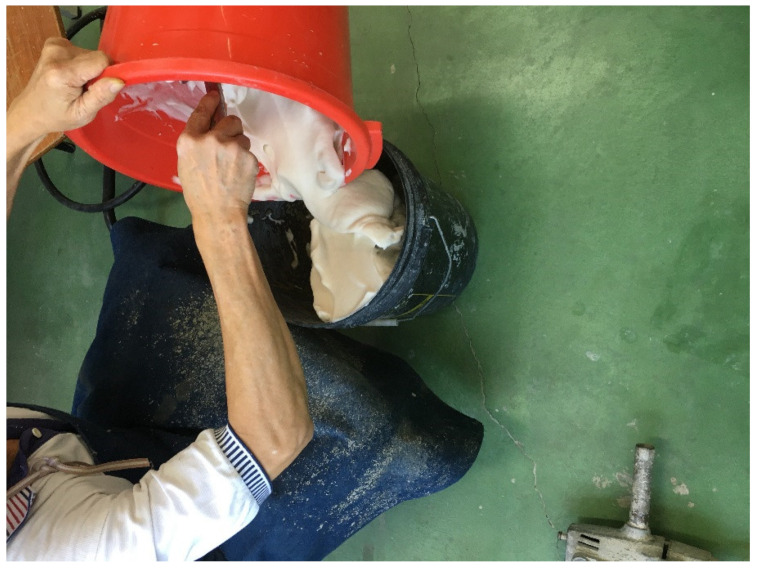
Mixing foam and slurry.

**Figure 4 materials-15-08128-f004:**
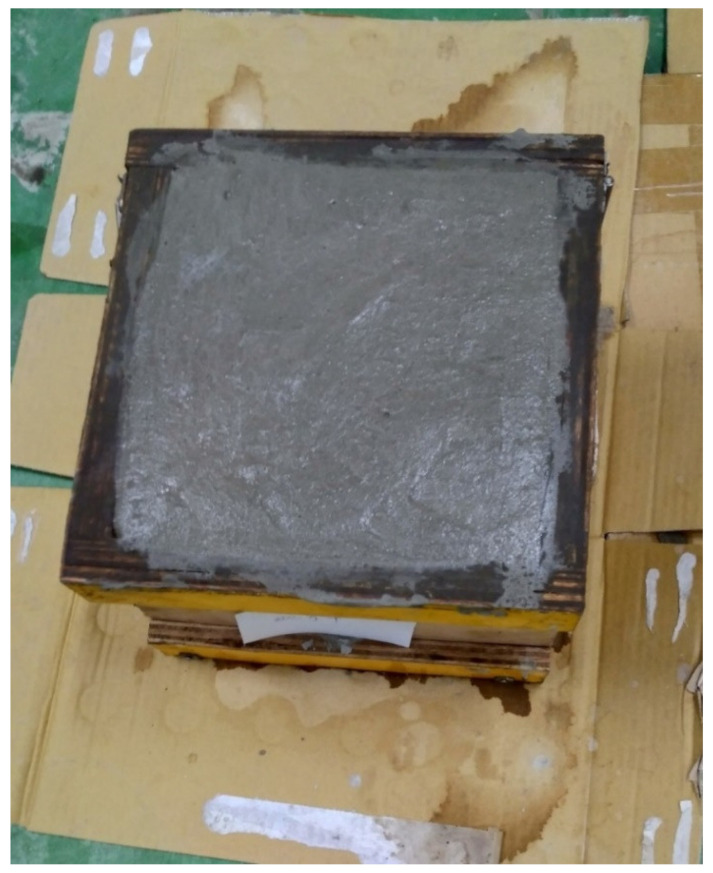
Foam concrete in a mold.

**Figure 5 materials-15-08128-f005:**
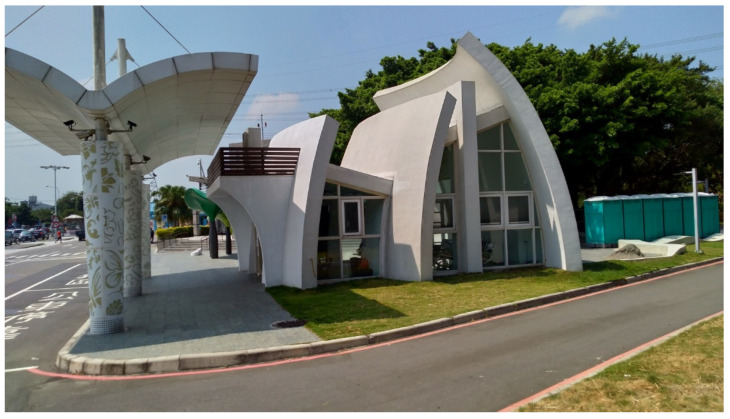
A shelter in Zhongli Service Area.

**Figure 6 materials-15-08128-f006:**
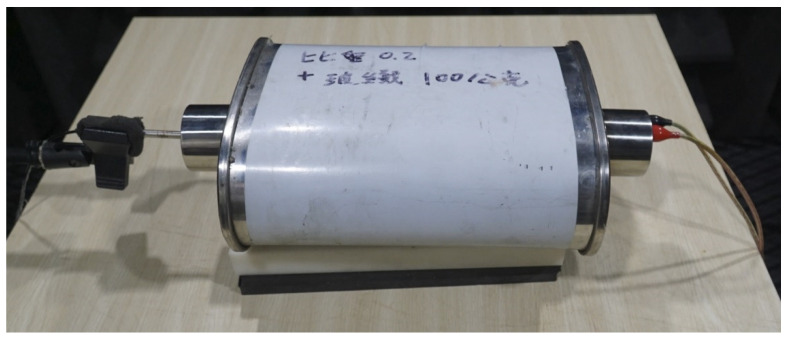
A foam concrete muffler at anechoic chamber.

**Figure 7 materials-15-08128-f007:**
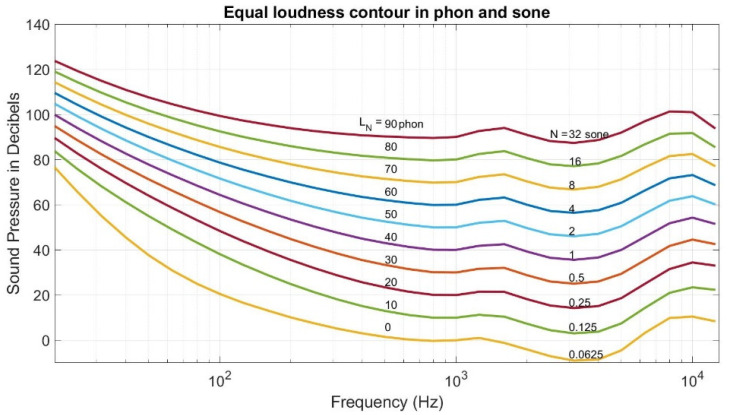
Equal loudness contour.

**Figure 8 materials-15-08128-f008:**
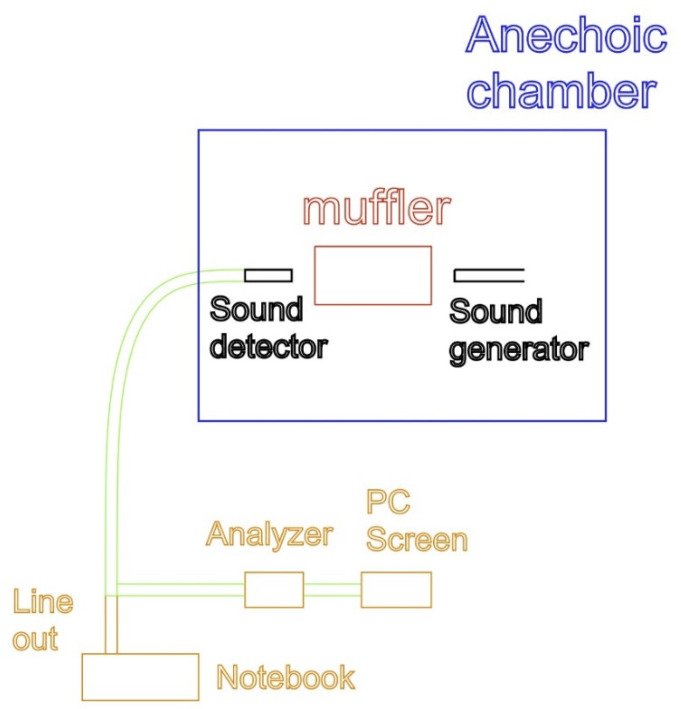
The experiment set up to retrieve sound.

**Figure 9 materials-15-08128-f009:**
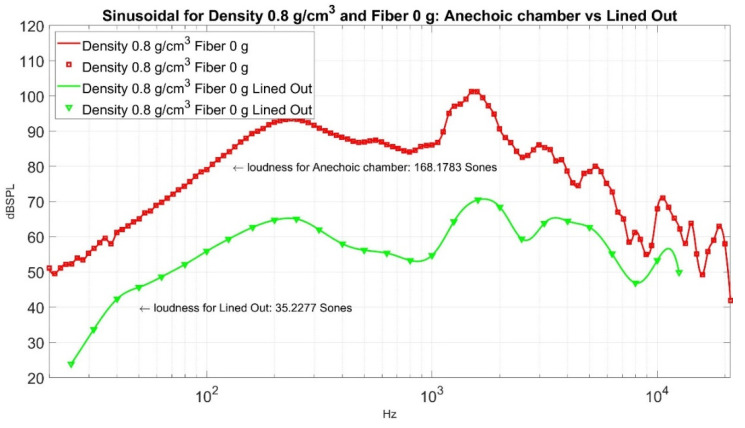
The sound pressure level curves for Density 0.8 g/cm^3^ and Fiber 0 g.

**Figure 10 materials-15-08128-f010:**
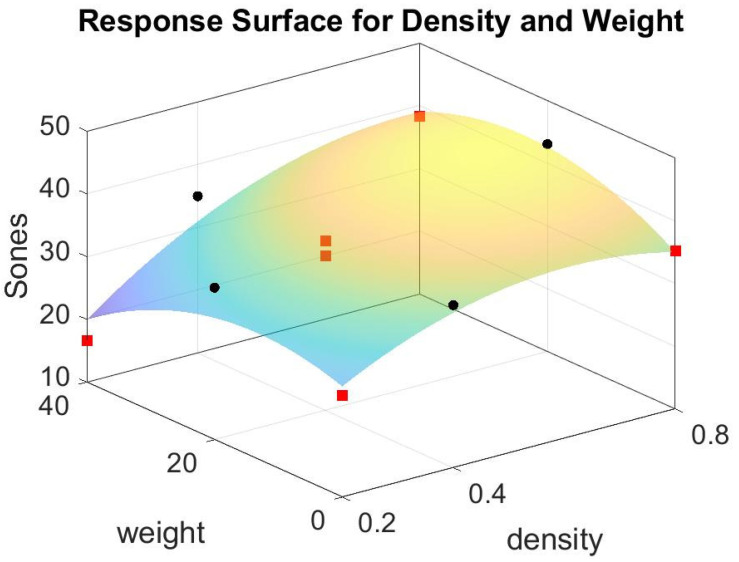
The response surface for ten data points in Table 4.

**Figure 11 materials-15-08128-f011:**
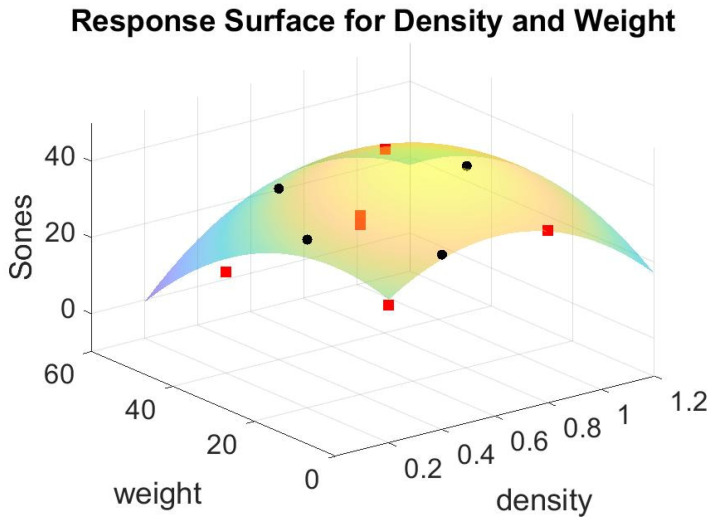
The response surface for ten data points in Table 4 in bigger range.

**Figure 12 materials-15-08128-f012:**
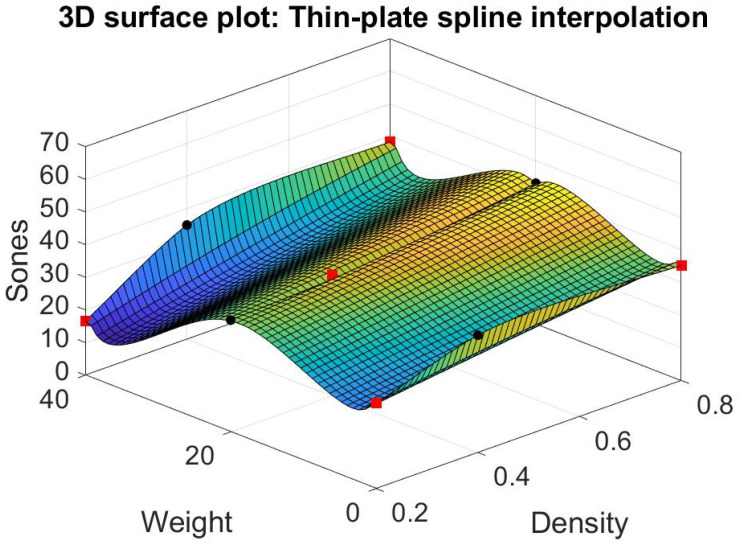
Thin-plate spline interpolation.

**Figure 13 materials-15-08128-f013:**
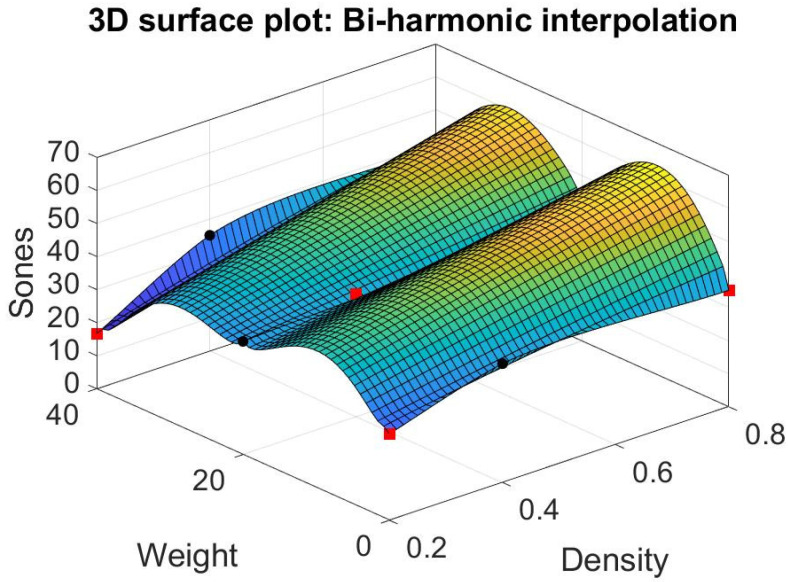
Bi-harmonic interpolation.

**Figure 14 materials-15-08128-f014:**
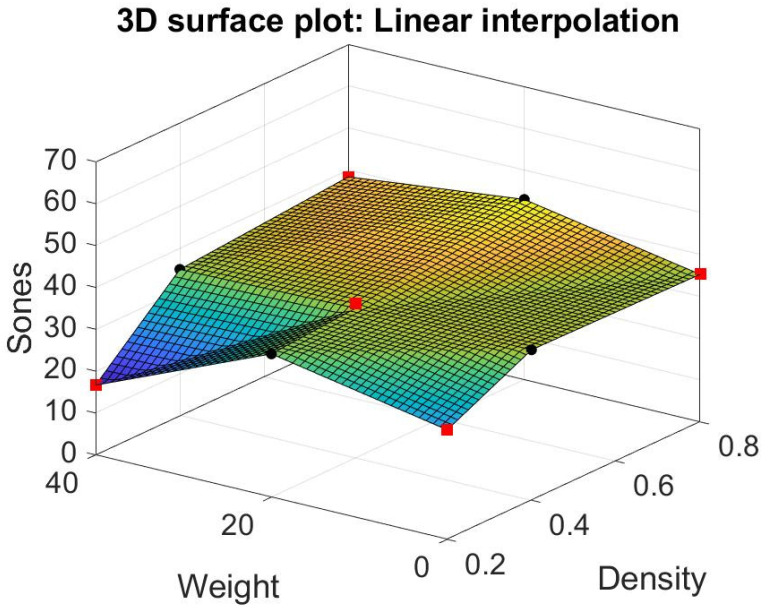
Linear interpolation.

**Figure 15 materials-15-08128-f015:**
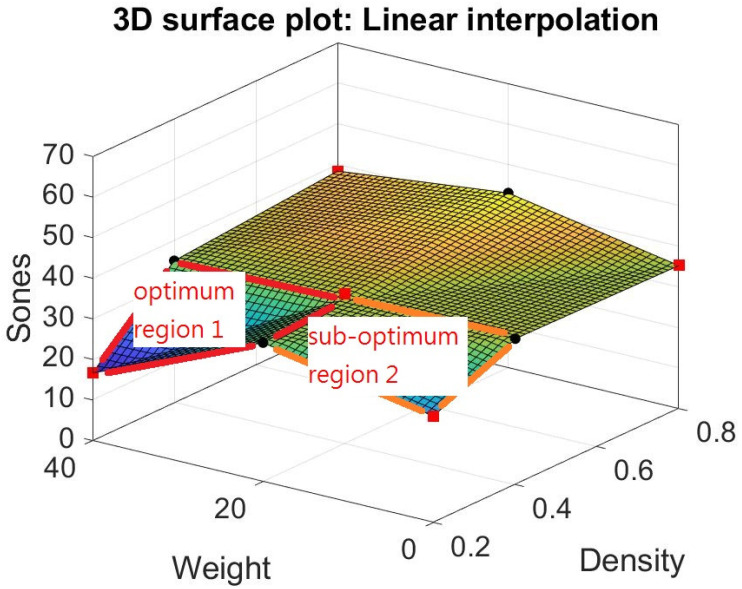
Two promising regions in linear interpolation.

**Figure 16 materials-15-08128-f016:**
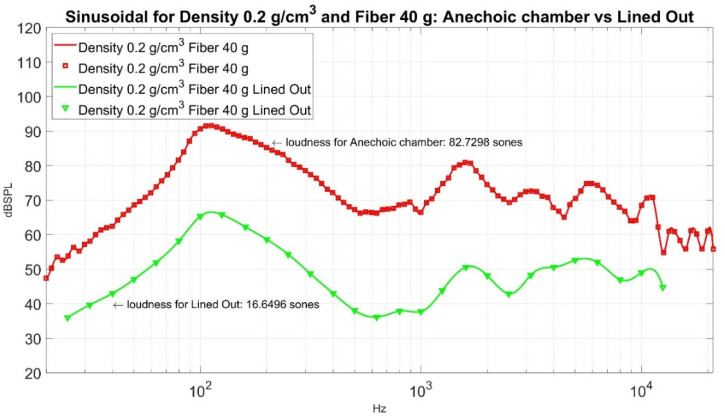
The sound pressure level curves for Density 0.2 g/cm^3^ and Fiber 40 g.

**Figure 17 materials-15-08128-f017:**
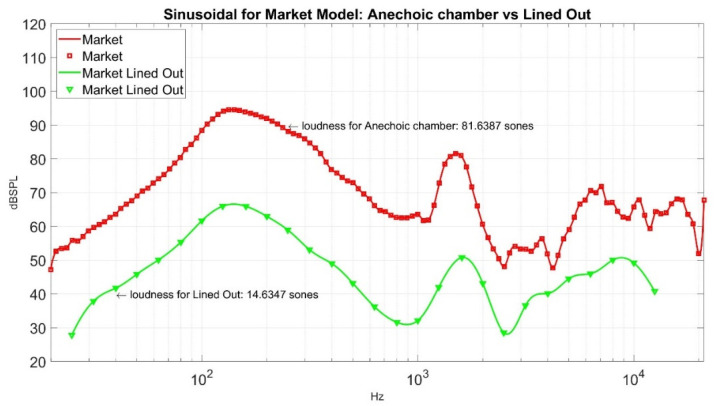
The sound pressure level curves for muffler in the market.

**Figure 18 materials-15-08128-f018:**
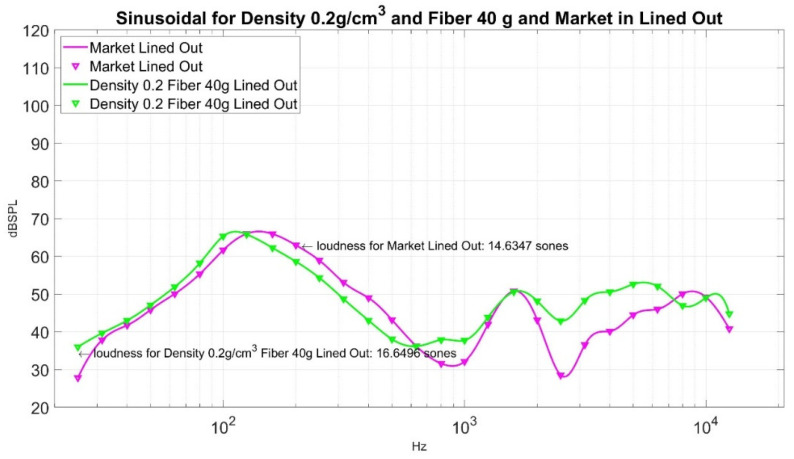
A comparison of sound pressure level curves for muffler in the market vs. Density 0.2 g/cm^3^ and Fiber 40 g in Lined Out.

**Figure 19 materials-15-08128-f019:**
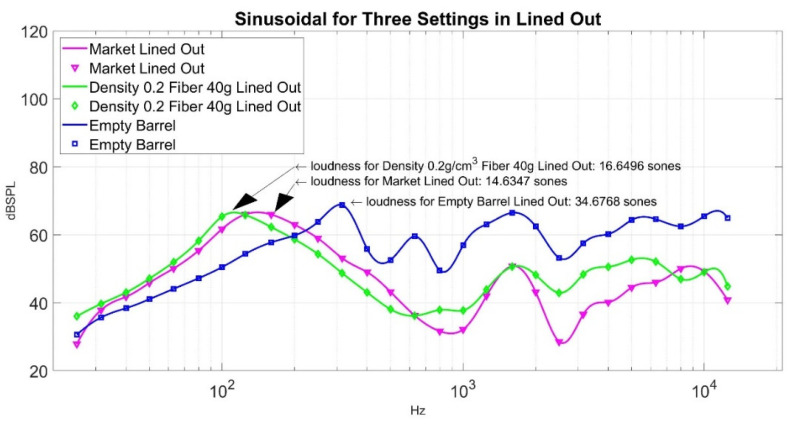
A comparison of sound pressure level curves for three settings in Lined Out.

**Figure 20 materials-15-08128-f020:**
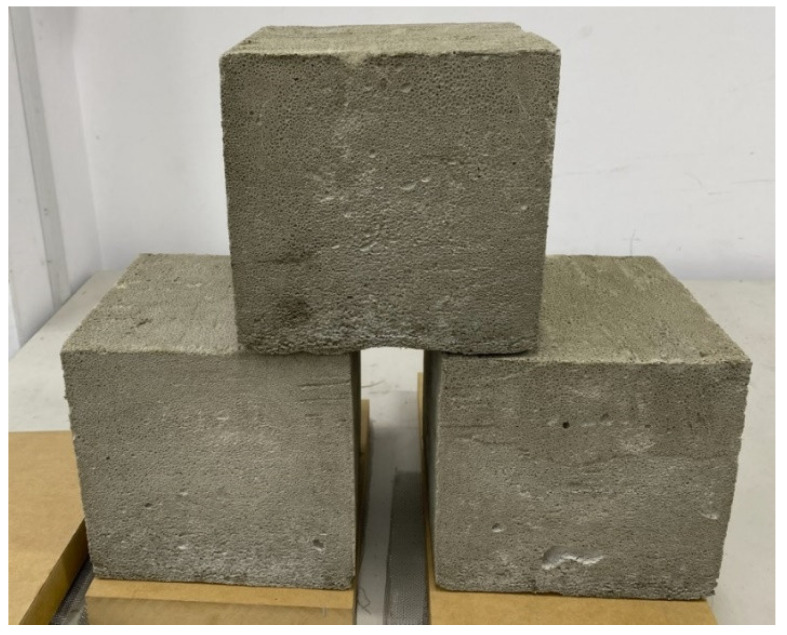
Three slab samples with a density of 0.2 g/cm^3^.

**Table 1 materials-15-08128-t001:** Comparison among car muffler devices.

Car Muffler Device	Strength	Weakness
Reactive muffler	No fiber/wool filling, environmentally friendly, suitable for low-frequency noise reduction; commercially available.	There are multiple partitions in the chamber structure. Complexity is induced.
Absorptive muffler	A simple chamber structure is suitable for high-frequency reduction; it is commercially available.	With fiber/wool filling could cause environmental hazards.
Hybrid muffler	It could reduce both low and high-frequency noise and is commercially available.	Manufacturing is costly.
MPP muffler	The theory of MPP on sound reduction has been established. No fiber/wool filling, environmentally friendly, and could reduce both low and high-frequency noise.	No mockup is available. The demonstration of the MPP muffler is too large to be used in the car muffler.

**Table 2 materials-15-08128-t002:** The chemical composition of the glass fiber (%).

SiO_2_	Al_2_O_3_	CaO	B_2_O_3_	MgO	Na_2_O + K_2_O	Fe_2_O_3_
52–56	12–16	15–25	5–10	0–6	0–1	0.05–0.4

**Table 3 materials-15-08128-t003:** The physical properties of the glass fiber.

Fiber Diameter	Temperature Duration
5–15 μ	650 °C

**Table 4 materials-15-08128-t004:** The two-way factorial design experiment layout and result.

Experiment No.	Density (g/cm^3^)	Weight (g)	Loudness Sensation (Sones)
1	0.2	0	26.1455
2	0.2	20	34.2075
3	0.2	40	16.6496
4	0.4	0	35.9190
5 (A)	0.4	20	36.9825
6 (B)	0.4	20	34.5275
7	0.4	40	34.9396
8	0.8	0	35.2277
9	0.8	20	43.0220
10	0.8	40	38.3396

**Table 5 materials-15-08128-t005:** The ANOVA of data in Table 4.

Analysis of Variance					
Source	DF	Adj SS	Adj MS	F-Value	*p*-Value
Model	5	404.876	80.975	4.51	0.085
Linear	2	265.988	132.994	7.42	0.045
Density	1	261.184	261.184	14.56	0.019
Fiber	1	4.803	4.803	0.27	0.632
Square	2	142.874	71.437	3.98	0.112
Density × Density	1	47.241	47.241	2.63	0.180
Fiber × Fiber	1	72.703	72.703	4.05	0.114
2-Way Interaction	1	35.389	35.389	1.97	0.233
Density × Fiber	1	35.389	35.389	1.97	0.233
Error	4	71.740	17.935		
Lack-of-Fit	3	78.726	22.909	7.60	0.259
Pure Error	1	3.014	3.014		
Total	9	476.616			

**Table 6 materials-15-08128-t006:** A comparison of the sound files.

Sound Reduction Conditions	Empty Barrel *	Market Muffler *	Green Muffler with Density of 0.2 g/cm^3^ and Fiber 40 g *
Sound wave from lined out	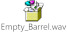	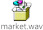	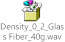

* All sound files can be found in Appendix A.

**Table 7 materials-15-08128-t007:** Formula for foam concrete with a density of 0.2 g/cm^3^.

Mixture	Items	Weight (g)
1	Lime	231
	Portland cement	807
	Calcium sulfoaluminate (CSA)	115
	Water 1	540
2	Foaming agent	38
	Water 2	657
3	Calcium chloride anhydrous	30
	Water 3	25

## Data Availability

Not applicable.

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
