# Peer review of "Using Response Surface for Searching the Nearly Optimal Parameters Combination of the Foam Concrete Muffler"

_materials, 2022, doi:10.3390/ma15228128_

Round 1
Reviewer 1 Report
Comments are listed below:
1. Strengthen the abstract section. Add the key conclusion of the works in the last two lines of the abstract section. Remove the unnecessary information.
2. Discuss the novelty of the work in respect of the application
3. There are numerous spelling and grammatical errors. Please revise the manuscript thoroughly. Sentences are also not complete and references are also cited in a rough manner.
4. Try to make a bridge between current and previously published work and specify the gap area and objective of the work. The introduction section is very poor: refer to following published work and write this paragraph in introduction section:
“ Wenchen et al. show the creep of aged concrete under extremely high sustained load. They also investigate the behavior of transverse reinforcement on the reinforced concrete creep during the loading and sustained load process. The data in this paper is precious because almost no relevant experiments can be found in previous research. The conclusion made in this research, such as a higher transverse reinforcement ratio can increase axial load capacity and reduce the concrete, especially in the early loading stage, provide great help for this research.”
- Ma, Wenchen, Ying Tian, Hailong Zhao, and Sarah L. Orton. "Time-Dependent Behavior of Reinforced Concrete Columns Subjected to High Sustained Loads." Journal of Structural Engineering 148, no. 10 (2022): 04022161.
5. Provide the image of the experimental setup with good quality. Also, add the image of the welded pipe produced.
6. The results are ok but the discussion section is very poor. It looks like a technical report instead of a technical article. Improve the discussion section and add more references in support of the results.
7. Shorten the length of the conclusion section.
8. The work is good, but the technical discussion and introduction section needs improvement. Paper can be accepted after following minor corrections.
Reviewer 2 Report
The authors have done a good work on this study, the methodology and results are clear and well discussed. However, some corrections in terms of English should be done.
1) the first sentence in page 2 should be refurmulated.
2) in table 1 expensive manufacturing instead of manufacturing is costly
3) The sentence "the operations are not environmentally friendly to workers"
should be modified.
4) in some sentences there is an excessif use of the preposition "of".
5) Do not use curves colors in the results analysis in case the copy is black and white.
Reviewer 3 Report
Review Letter
Using Response Surface for Searching the Nearly Optimal Parameters Combination of the Foam Concrete Muffler
Review of the Manuscript with paper ID of materials-2014103
This present research aims to study the feasibility of the composite as a raw material for making the filling material in the absorptive muffler. It assesses the effectiveness of using a varying density of lightweight concrete and amount of glass fiber on the sound reduction performance in the absorptive muffler. The publication of this article needs to consider the following comments completely:
1. What is the difference between the flowchart of the current approach and that of [19]?
2. In order to reduce the noise, which strategy is used by the authors?
3. Please remake the novelty of the text more strongly compared to previous approaches.
4. Please explore that how equation (2) is developed?
5. Based on the results of Fig.20, what subjects can be remarked?
6. Conclusion part of the study should be summarized. Please focus on the main achievements of the approach.
7. The transmission loss (TL) coefficient should be confirmed based on https://doi.org/10.1177/1099636221993891 and https://doi.org/10.1016/j.ast.2020.106141.
8. Please explain that why there is a negligible difference between the market muffler and green muffler sound waves?
Reviewer 4 Report
This manuscript "Using Response Surface for Searching the Nearly Optimal Parameters Combination of the Foam Concrete Muffler" written by Teng-Hsuan Lin, Jyhjeng Deng and Yi-Ching Chen represents a good contribution for your valuable Journal.
However, before the Editor makes a decision, I suggest that the authors must take into account the following corrections:
1. The authors would explain clearly in the abstract what is the novelty of the proposed method and what is the added value in this article. Conclusions would be more carefully rewritten, summarizing what has been learned and why it is interesting and useful.
2. The author should check typing errors throughout the manuscript. English style should also be improved.
3. The proposed methodology, contains many well know equations, which are already published. The methodology sections should be reduced and appropriate references should be added. The authors should keep only their new contributions.
4. References are suggestive. I am convinced that it is useful for the manuscript if will be included in the References section following papers with the same topics or using similar procedures, ex.: Strength Parameters of Foamed Geopolymer Reinforced with GFRP Mesh. Materials 2021, 14, 689. DOI:10.3390/ma14030689; Internal structure influence on the impact strength and dynamic fracture toughness of hybrid polymer matrix composites integrated with elastomer layers, Composite Structures, 258(2), 113375, (2021) DOI:10.1016/j.compstruct.2020.113375; Mechanics of Elastic Composites, Chapman & Hall/ CRC Press, U.S.A, 708 pp., (2003).
For these reasons, I recommend the acceptance of this manuscript for publication after Major Revision.
Round 2
Reviewer 1 Report
The authors have only superficially reformed the manuscript, focusing mostly on the changes related to grammatical, style and structural errors/modifications. The introduction still lacks the scientific soundness required to explain the necessity for the performed research and experiments as well as it provides insufficient references to certain statements from the authors. Sadly, the remaining manuscript has not been altered and provides no new changes that would improve the scientific quality and soundness of the proposed manuscript. The authors have disregarded all the scientific questions, queries and notions that have been provided to them. The manuscript is still in a considerable disarray and provides no new findings that would be important for the research field. Furthermore, the authors have not modified the discussion from the scientific point and have left all the scientific messages the same (even those that are misleading, incorrect and/or incomplete). As such, I still cannot recommend the manuscript for publication.
Reviewer 4 Report
Accept in the reviewed form.
Round 3
Reviewer 1 Report
Accept